# Trends and determinants of ever having tested for HIV among youth and adults in South Africa from 2005–2017: Results from four repeated cross-sectional nationally representative household-based HIV prevalence, incidence, and behaviour surveys

**Sean Jooste**[1]*, **Musawenkosi Mabaso**[1], **Myra Taylor**[2], **Alicia North**[1], **Rebecca Tadokera**[1,3], **Leickness Simbayi**[4,5]

1 Social Aspects of Public Health Research Programme, Human Sciences Research Council, Cape Town, South Africa, 2 School of Nursing and Public Health, University of KwaZulu-Natal, Durban, South Africa, 3 Division of Molecular Biology and Human Genetics, NRF/DST Centre of Excellence for Biomedical Tuberculosis Research, South African Medical Research Council Centre for Tuberculosis Research, Faculty of Medicine and Health Sciences, Stellenbosch University, Cape Town, South Africa, 4 Deputy CEO for Research, Human Sciences Research Council, Cape Town, South Africa, 5 Department of Psychiatry & Mental Health, University of Cape Town, Cape Town, South Africa

* sjooste@hsrc.ac.za

## Abstract

### Background

HIV testing contributes to the prevention and control of the HIV epidemic in the general population. South Africa has made strides to improve HIV testing towards reaching the first of the UNAIDS 90–90–90 targets by 2020. However, to date no nationally representative analysis has examined temporal trends and factors associated with HIV testing among youth and adults in the country.

### Aim

This study aimed to examine the trends and associations with ever having tested for HIV among youth and adults aged 15 years and older in South Africa using the 2005, 2008, 2012 and 2017 nationally representative population-based household surveys.

### Methods

The analysis of the data collected used multi-stage stratified cluster randomised cross-sectional design. P-trend chi-squared test was used to identify any significant changes over the four study periods. Bivariate and multivariate logistic regression analysis was conducted to determine factors associated with HIV testing in each of the survey periods.

### Results

Ever having tested for HIV increased substantially from 2005 (30.6%, n = 16 112), 2008 (50.4%, n = 13 084), 2012 (65.5%, n = 26 381), to 2017 (75.2%, n = 23 190). Those aged 50

**Data Availability Statement:** All relevant data are within the paper and its Supporting Information files.

**Funding:** The study was funded by the President's Emergency Plan for AIDS Relief (PEPFAR) under the terms of 5U2GGH000570. The funders had no role in study design, data collection and analysis, decision to publish, or preparation of the manuscript.

**Competing interests:** The authors have declared that no competing interests exist.

years and older were significantly less likely to ever have tested for HIV than those aged 25–49 years. Those residing in rural areas were significantly less likely to have tested for HIV as compared to people from urban areas.

There was a change in HIV testing among race groups with Whites, Coloureds and Indian/Asians testing more in 2005 and 2008 and Black Africans in 2017. Marriage, education and employment were significantly associated with increased likelihood of ever testing for HIV. Those who provided a blood specimen for laboratory HIV testing in the survey rounds and were found to have tested positive were more likely to have ever tested for HIV previously.

## Conclusion

The results show that overall there has been an increase in ever having an HIV test in the South African population over time. The findings also suggest that for South Africa to close the testing gap and reach the first of the UNAIDS 90–90–90 targets by 2020, targeted programmes aimed at increasing access and utilization of HIV testing in young people, males, those not married, the less educated, unemployed and those residing in rural areas of South Africa should be prioritised.

## Introduction

HIV testing is a critical step in the HIV treatment cascade, which includes diagnosis, linkage to care, engagement in care, initiation of antiretroviral therapy, retention in care, and sustained viral load suppression [1]. HIV testing also contributes to the prevention and control of the HIV epidemic in the general population, since people diagnosed with HIV can make decisions that potentially lower their risk of HIV transmission and re-infection, while those who test negative can make informed decisions to protect themselves from getting infected [2].

Evidence shows that decision-making and practices related to HIV-testing could be influenced by several factors including accurate knowledge about HIV transmission, perceived risk of HIV infections, attitudes and perceptions of HIV-testing services, and previous history and experiences of HIV-testing [3]. HIV testing uptake may also be influenced by individual level factors such as gender, age and marital status, and socio-economic characteristics such as urban or rural residence, education attainment, and employment status among others [3, 4]. Understanding these associations is important for making effective interventions aimed at containing the HIV epidemic, particularly as countries aim to attain the UNADS 90–90–90 targets. These targets are aimed to increase knowledge of HIV-positive status, initiation of antiretroviral therapy (ART) and viral suppression by 2020 [1, 5, 6]. Changes in HIV testing overtime has implications for the UNAIDS targets and for evaluating the impact of national policies [7].

In South Africa, policy initiatives aligned to meet UNAIDS 90-90-90 targets have an impact on HIV testing [8]. The Government has embarked on a deliberate effort to scale up HIV testing services (HTS) by increasing availability of quality HTS and its uptake in all public health facilities [9]. Scaling up of HIV testing HIV testing has the potential to affect the first '90' of the UNAIDS 90-90-90 targets.

Due to the high burden of HIV overtime, the country has experienced demographic, socio-economic and behavioural change because of the epidemiological transition [10, 11]. It is

therefore important to monitor changes in HIV testing and important indicators in order to inform policy. This study examined temporal trends and factors associated with HIV testing in the South Africa population aged 15 years and older using data from the 2005, 2008, 2012, and 2017 nationally representative population-based household surveys.

## Methods

All youth and adults who agreed to participate were required to provide written or verbal (where respondent was illiterate). Written Informed consent for participating in the survey and for the collection of the dried blood spot specimens were obtained or participants 18 years and older. For those less than 18 years informed consent was obtained from parents or guardians and assent obtained from the participants. Where people could not write verbal consent was obtained and fieldwork supervisor signed as witness. The four survey protocols were approved by the Human Sciences Research Council Research Ethics Committee (REC approval numbers: 5/24/06/04; 2/23/10/07; 5/17/11/10 and 4/18/11/15).

### Data

Data from the 2005, 2008, 2012 and 2017 South African national household-based HIV Prevalence, Incidence and Behaviour surveys were used for this study [12–15]. The methodology for these surveys has been the same across the different survey waves, hence allowing this comparison of temporal trends [12–15]. The surveys used a multi-stage stratified cluster randomised cross-sectional design. In each survey a systematic probability sample of 15 households was randomly chosen from 1 000 enumeration areas (EAs) which were randomly selected from 86 000 EAs based on the national sampling frame released by Statistics South Africa in 2001 and updated in 2011 [16–18]. The sampling of EAs was stratified by province and locality type (urban formal, urban informal, rural formal—including commercial farms and rural informal localities).

### Study participants

In the 2005 and 2008 surveys, in each household a maximum of three people were selected randomly to participate in the study, each representing the 2–14 years, 15–24 years and 25 years and older age groups. In the 2012 and 2017 surveys, all household members were eligible to participate in the survey. In all the surveys age appropriate questionnaires were administered to solicit information on socio-demographic characteristics, sexual practices and behaviour, knowledge, attitudes and perceptions, testing of tuberculosis and HIV, exposure to HIV media campaigns, alcohol and substance use and general health related characteristics. The current analysis focused on individuals aged 15 years and older who responded to the question on ever tested for HIV.

### HIV testing

Dried blood spots' (DBS) specimens were collected from consenting individuals for HIV testing. Samples were tested for HIV using an enzyme immunoassay (EIA) (Vironostika HIV Uni-Form II plus O, Biomeriux, Boxtel, The Netherlands), and samples which tested positive were retested using a second EIA (Advia Centaur XP, Siemens Medical Solutions Diagnostics, Tarrytown, New York, USA). Any samples with discordant results on the first two EIAs were tested with a third EIA (Roche Elecys 2010 HIV Combi, Roche Diagnostics, Mannheim, Germany).

## Measures

### Primary outcome

In all four surveys, participants who responded to the question "Have you ever been tested for HIV?" are included in the analysis. The response was dichotomized for the primary outcome (yes = 1 and no = 0).

### Explanatory variables

Covariate socio-demographic variables such as age, grouped into 15–24 years, 25–49 years, 50 years and older, sex (male, female), race (Black Africans, White, Coloured, Indian/Asian), current marital status (not married, married), level of education (no education, primary, secondary, tertiary), employment status (unemployed, employed), and locality type (urban areas, rural informal, rural formal areas) were included.

Other covariates included HIV behavioural variables such as ever had sexual intercourse (no, yes), age of sexual debut (had sex before the age of 15 years, had sex aged 15 years and older), age of sexual partner (partner more than five years younger, partner within five years of age, partner more than five years older), multiple sexual partners in the last 12 months (one partner, two or more partners), condom use at last sex (yes, no), the Alcohol Abuse Disorder Identification Test (AUDIT) score (abstainers, low risk (with scores ranging from 1–7), risky/hazardous level (8–15), high risk/harmful (16–19), very high risk (20+), [19]. Accurate knowledge about preventing the sexual transmission of HIV was based on responses to five prompted questions; 'Can a person reduce the risk of getting HIV by using a condom every time they have sex?' 'Can a person reduce the risk of HIV by having fewer sexual partners?', 'Can AIDS be cured?' 'Can a person get HIV by sharing food with someone who is infected?' 'Can a healthy-looking person have HIV? Accurate knowledge of preventing the sexual transmission of HIV and rejection of misconceptions about HIV transmission was scored 1 'yes' if all five items were correctly answered, whereas if they answered any incorrectly, they scored 0 'no'. Self-perceived risk of HIV infection (yes, no) and antibody detected HIV status (HIV positive, HIV negative) were also included.

## Statistical analysis

Descriptive statistics of socio-demographic variables and HIV risk behaviours by HIV testing were generated for each of the study waves, using frequencies and proportions. P-trend chi-squared test was used to identify any significant change over the four study periods. Percentage differences in HIV testing between the 2005 and 2017 surveys were also calculated. Bivariate logistic regression analysis was used to assess factors associated with having ever been tested for HIV for each study wave. All statistically significant variables were entered into a final multivariate logistic regression model. Crude Odds ratios (OR) and adjusted (aOR) for the bivariate and multivariate models, with 95% Confidence intervals (CI) and a p-value ≤ 0.05 were considered statistically significant. Separate models were fitted for each of the survey waves. Sample weights were introduced to all models to account for the complex survey design and non-response using the 'svy' command. Statistical analyses were done using Stata statistical software, Release 15.0 (College Station, TX: Stata Corporation).

## Results

### Trends in ever having tested for HIV in 2005–2017

Table 1 shows trends in ever having tested for HIV by socio-demographic characteristics among youth and adults aged 15 years and older. Overall the percentage of those ever been

**Table 1. Trend in ever having tested for HIV by socio-demographic characteristics of the study participants age 15 years and older in South Africa by survey period from 2005–2017.**

| Variables | 2005 | | | 2008 | | | 2012 | | | 2017 | | | Percentage point increase | p-value* |
|---|---|---|---|---|---|---|---|---|---|---|---|---|---|---|
| | n | % | 95% CI | n | % | 95% CI | n | % | 95% CI | n | % | 95% CI | | |
| **Age categories** | | | | | | | | | | | | | | |
| Total | 16 112 | 30.6 | 29.1–32.1 | 13 084 | 50.4 | 48.9–51.8 | 26 381 | 65.5 | 64.2–66.7 | 23 190 | 75.2 | 74.0–76.4 | 44.6 | <0.001 |
| 15–24 years | 5 615 | 19.3 | 17.7–20.9 | 4 192 | 37.3 | 35.1–39.6 | 7 121 | 50.6 | 48.5–52.7 | 5 921 | 58.8 | 56.6–61.1 | 39.5 | <0.001 |
| 25–49 years | 6 764 | 43.1 | 40.8–45.5 | 5 606 | 65.1 | 63.0–67.2 | 11 553 | 78.2 | 76.6–79.8 | 10 674 | 85.0 | 83.6–86.2 | 41.9 | <0.001 |
| 50+ years | 3 733 | 18.3 | 16.4–20.4 | 3 286 | 34.1 | 31.6–36.6 | 7 707 | 54.8 | 52.7–57.0 | 6 595 | 69.7 | 68.0–71.4 | 51.4 | <0.001 |
| **Sex of respondent** | | | | | | | | | | | | | | |
| Male | 6 193 | 27.6 | 25.5–29.8 | 5 193 | 44.2 | 42.0–46.4 | 11 403 | 59.0 | 57.2–60.8 | 9 762 | 70.9 | 69.2–72.5 | 43.3 | <0.001 |
| Female | 9 919 | 33.0 | 31.3–34.7 | 7891 | 55.8 | 54.1–57.4 | 14 978 | 71.5 | 70.1–72.9 | 13 428 | 79.3 | 78.0–80.5 | 46.3 | <0.001 |
| **Race of respondent** | | | | | | | | | | | | | | |
| Black African | 9 515 | 26.2 | 24.6–27.8 | 7 844 | 49.0 | 47.3–50.7 | 15 166 | 65.8 | 64.3–67.2 | 15 255 | 76.5 | 75.1–77.9 | 50.3 | <0.001 |
| White | 1 888 | 53.2 | 48.5–57.9 | 1 570 | 59.0 | 55.4–62.5 | 2 823 | 62.7 | 58.8–66.4 | 1 634 | 69.4 | 65.9–72.8 | 16.2 | <0.001 |
| Coloured | 2 949 | 36.2 | 33.1–39.4 | 2 348 | 51.0 | 48.1–54.0 | 4 911 | 67.8 | 65.4–70.1 | 4 182 | 73.8 | 71.7–75.7 | 37.6 | <0.001 |
| Indian/Asian | 1 728 | 44.7 | 39.6–49.8 | 1 294 | 51.6 | 46.0–57.2 | 3 419 | 60.6 | 55.7–65.3 | 2 119 | 61.8 | 56.7–66.7 | 17.1 | <0.001 |
| **Marital status** | | | | | | | | | | | | | | |
| Not married | 10 160 | 25.5 | 24.0–27.1 | 8 392 | 47.4 | 45.6–49.1 | 16 707 | 62.4 | 60.8–64.0 | 7 391 | 81.1 | 79.6–82.6 | 55.6 | <0.001 |
| Married | 5 917 | 39.1 | 36.6–41.8 | 4 649 | 56.1 | 53.8–58.4 | 9 276 | 73.2 | 71.3–75.0 | 15 794 | 72.8 | 71.4–74.2 | 33.7 | <0.001 |
| **Level of education** | | | | | | | | | | | | | | |
| No education/ Primary | 4 537 | 18.4 | 16.4–20.5 | 3 291 | 35.8 | 33.4–38.3 | 4 285 | 58.4 | 56.0–60.7 | 3 626 | 71.9 | 69.8–74.0 | 53.5 | <0.001 |
| Secondary | 9 955 | 31.7 | 29.9–33.5 | 8 384 | 52.3 | 50.5–54.1 | 15 883 | 66.2 | 64.6–67.8 | 11 266 | 80.4 | 79.0–81.7 | 48.7 | <0.001 |
| Tertiary | 1 571 | 61.4 | 57.6–65.1 | 1 330 | 72.7 | 68.7–76.4 | 2 248 | 81.8 | 78.2–84.8 | 2 471 | 85.2 | 82.9–87.3 | 23.8 | <0.001 |
| **Employment status** | | | | | | | | | | | | | | |
| Unemployed | 10 822 | 23.2 | 21.9–24.7 | 8 163 | 43.3 | 41.8–44.9 | 14 298 | 61.3 | 59.7–62.9 | 14 796 | 70.7 | 69.2–72.0 | 47.5 | <0.001 |
| Employed | 5 223 | 46.0 | 43.0–49.1 | 4 854 | 62.3 | 59.8–64.7 | 9 658 | 75.2 | 73.2–77.0 | 8 070 | 83.7 | 82.1–85.2 | 37.7 | <0.001 |
| **Locality type** | | | | | | | | | | | | | | |
| Urban areas | 11 011 | 38.9 | 36.9–40.9 | 9 370 | 53.7 | 51.9–55.5 | 18 271 | 68.0 | 66.4–69.6 | 15 082 | 77.4 | 76.0–78.7 | 38.5 | <0.001 |
| Rural informal areas | 3 682 | 19.4 | 17.3–21.5 | 2 837 | 43.1 | 40.6–45.6 | 5 602 | 61.6 | 59.6–63.5 | 5 432 | 70.3 | 68.0–72.4 | 50.9 | <0.001 |
| Rural formal areas | 1 419 | 21.0 | 16.7–26.1 | 877 | 50.4 | 45.6–55.1 | 2 508 | 63.0 | 57.4–68.2 | 2 676 | 71.2 | 67.2–74.9 | 50.2 | <0.001 |

* p-value ≤ 0.05 was considered statistically significant

tested for HIV increased from 30.6% in 2005 to 75.2% in 2017 (a 44.6% point increase). There was a significant increase in the percentages of respondents who had ever been tested for HIV across all socio-demographic characteristics from 2005 to 2017 (p<0.001 for all variables). The largest increase was among those aged 50 and older changing from 18.3% in 2005 to 69.7% in 2017 (a 51.4% point increase), Black Africans from 26.2% in 2005 to 76.5% in 2017 (a 50.3% point increase), those with no education or who had only attained primary school education from 18.4% in 2005 to 71.9% in 2017 (a 53.5% point increase), those unemployed from 23.2% in 2005 to 70.7% in 2017 (a 47.5% point increase), and those residing in rural informal areas from 19.4% to 70.3% in 2017 (a 50.9% point increase).

Table 2 shows trends in ever having tested for HIV by risk behaviour characteristics among youth and adults aged 15 years and older. There was a significant increase in the percentages of respondents who had ever been tested for HIV across all HIV risk behaviour characteristics in 2005 to 2017 (p<0.001 for all variables). The largest increase was among those who ever had sex from 34.6% in 2005 to 81.1% in 2017 (a 46.5% point increase), those with sexual partners 5 years and older than themselves, from 29.7% in 2005 to 89.9% in 2017 (a 60.2% point increase), those with two or more sexual partners from 35.5% in 2005 to 83.0% in 2017 (a 47.5% point increase), among those who reported no alcohol consumption from 27.0% in 2005 to 73.6 in 2017 (a 46.6% point increase) and among those who tested HIV positive in the survey, from 36.7% in 2005 to 87.1% in 2017 (a 50.4% point increase).

## Determinants of ever having tested for HIV in 2005–2017

All variables in the bivariate logistic regression analysis were statistically significant and therefore they were all controlled for in the multivariate logistic regression analysis. Fig 1 shows coefficient plots of the final multiple logistic regression models of determinants of ever being tested for HIV in each survey wave. The increased likelihood of ever being tested for HIV was significantly associated with those aged 25–49 years rather than those aged 15–24 years, in 2005 [aOR = 1.33 (95% CI: 1.06–1.66), p = 0.013], 2008 [aOR = 2.08 (95% CI: 1.75–2.47), p<0.001], 2012 [aOR = 2.5 (95% CI: 2.1–3.0) p<0.001], and 2017 [aOR = 2.5 (95% CI: 2.1–3.0), p<0.001]. Females were significantly more likely to have ever tested for HIV than males, in 2005 [aOR = 1.74 (95% CI: 1.44–2.10), p<0.001], 2008 [aOR = 1.83 (95% CI: 1.59–2.11), p<0.001], 2012 [aOR = 2.3 (95% CI: 2.0–2.6), p<0.001], and 2017 [aOR = 2.05 (95% CI: 1.88–2.23), p<0.001]. Whites were significantly more likely to have ever tested for HIV than Black Africans, in 2005 [aOR = 2.64 (95% CI: 1.99–3.49), p<0.001] and in 2008 [aOR = 1.31 (95% CI: 1.02–1.68), p = 0.031]. Similarly, Indians [aOR = 1.57 (95% CI: 1.2–2.1), p<0.001], and Coloureds [aOR = 1.61 (95% CI: 1.3–2.0), p<0.001], were significantly more likely to ever have tested for HIV in 2005 than Black Africans. Married respondents were significantly more likely to have ever tested for HIV than those unmarried: in 2005 [aOR = 1.24 (95% CI: 1.04–1.47), p = 0.018], in 2008 [aOR = 1.23 (95% CI: 1.06–1.44), p = 0.008], in 2012 [aOR = 1.60 (95% CI: 1.3–1.9), p<0.001], and in 2017 [aOR = 1.58 (95% CI: 1.41–1.76), p<0.001].

The increased likelihood of ever being tested for HIV was also significantly associated with attainment of a secondary school level of education compared to those who had no education or those who only had attained primary school education, in 2005 [aOR = 1.56 (95% CI:1.26–1.93, p<0.001], 2008 [aOR = 1.80 (95% CI:1.52–2.12), p<0.001], 2012 [aOR = 1.5 (95% CI:1.3–1.7), p<0.001], and 2017 [aOR = 1.76 (95% CI: 1.58–1.96), p<0.001]. Tertiary level education had higher likelihood of ever being tested for HIV, in 2005 [aOR = 3.74 (95% CI:2.74–5.12), p<0.001], 2008 [aOR = 3.74 (95% CI:2.74–5.11), p<0.001], 2012 [aOR = 2.7 (95% CI:1.9–4.0), p<0.001], and 2017 [aOR = 2.66 (95% CI:2.21–3.20), p<0.001] compared to the same referent. Employed respondents were significantly more likely to have ever tested for

**Table 2.** Trend in every having tested for HIV by HIV risk behaviour characteristics of the study participants age 15 years and older in South Africa by survey period from 2005–2017.

| Variables | 2005 | | | 2008 | | | 2012 | | | 2017 | | | Percentage point increase | p-value[*] |
|---|---|---|---|---|---|---|---|---|---|---|---|---|---|---|
| | n | % | 95% CI | n | % | 95% CI | n | % | 95% CI | n | % | 95% CI | | |
| **Sexual activity** | | | | | | | | | | | | | | |
| Never had sex | 2 596 | 8.3 | 6.4–10.6 | 1 910 | 15.8 | 13.3–18.7 | 3 317 | 28.4 | 25.6–31.4 | 3 655 | 43.4 | 40.5–46.3 | 35.1 | <0.001 |
| Had sex | 13 148 | 34.6 | 33.0–36.3 | 9 586 | 55.2 | 53.7–56.8 | 22 360 | 71.0 | 69.7–72.3 | 18 319 | 81.1 | 80.0–82.2 | 46.5 | <0.001 |
| **Sexual debut** | | | | | | | | | | | | | | |
| Sex before aged 15 | 248 | 25.4 | 19.0–32.9 | 192 | 44.5 | 34.9–54.4 | 391 | 57.9 | 50.3–65.2 | 387 | 66.1 | 59.6–72.0 | 40.7 | <0.001 |
| Sex at aged 15 and older | 5 367 | 19.0 | 17.3–20.7 | 3 911 | 36.8 | 34.5–39.2 | 6 696 | 50.0 | 47.9–52.2 | 5 513 | 58.1 | 55.8–60.4 | 39.1 | <0.001 |
| **Age of sexual partner** | | | | | | | | | | | | | | |
| Partner more than 5 years younger | 1 624 | 36.2 | 32.6–40.1 | 1 138 | 52.1 | 47.5–56.6 | 2 659 | 72.8 | 69.7–75.7 | 2 21 | 81.6 | 79.1–83.9 | 45.4 | <0.001 |
| Partner within five years | 5 329 | 39.4 | 36.9–42.1 | 3 997 | 60.1 | 57.8–62.4 | 9 739 | 73.2 | 71.4–74.9 | 6 874 | 84.1 | 82.7–85.5 | 44.7 | <0.001 |
| Partner more than 5 years older | 5 943 | 29.7 | 27.8–31.8 | 1 448 | 65.9 | 62.2–69.5 | 3 241 | 86.3 | 84.3–88.1 | 2 728 | 89.9 | 88.3–91.3 | 60.2 | <0.001 |
| **Number of sexual partners in the past 12 months** | | | | | | | | | | | | | | |
| 1 sexual partner | 8 444 | 39.5 | 37.5–41.6 | 648 | 47.8 | 41.8–54.0 | 14 183 | 77.0 | 75.7–78.3 | 10 845 | 84.9 | 83.8–86.0 | 45.4 | <0.001 |
| 2 or more sexual partners | 674 | 35.5 | 29.6–41.8 | 7 159 | 61.8 | 60.0–63.6 | 1 487 | 67.8 | 64.0–71.4 | 1 049 | 83 | 79.8–85.9 | 47.5 | <0.001 |
| **Condom use at last sex in the past 12 months** | | | | | | | | | | | | | | |
| No condom use | 6 215 | 39.7 | 37.3–42.2 | 4 900 | 59.9 | 57.5–62.1 | 10 663 | 75.7 | 74.0–77.4 | 7 657 | 84.6 | 83.2–85.9 | 44.9 | <0.001 |
| Yes condom use | 2 961 | 38.2 | 35.5–41.0 | 2 887 | 61.0 | 58.4–63.6 | 4 707 | 76.0 | 73.9–78.0 | 4 144 | 85 | 83.4–86.4 | 46.8 | <0.001 |
| **AUDIT Score** | | | | | | | | | | | | | | |
| Abstainers | 10 979 | 27.0 | 25.4–28.6 | 8 892 | 48.0 | 46.4–49.7 | 14 638 | 64.3 | 62.8–65.8 | 15 168 | 73.6 | 72.0–75.1 | 46.6 | <0.001 |
| Low risk (1–7) | 2 905 | 39.4 | 36.3–42.6 | 2 826 | 59.6 | 56.7–62.5 | 6 326 | 68.3 | 65.8–70.6 | 4 067 | 77.6 | 75.5–79.6 | 38.2 | <0.001 |
| Risky/hazardous level (8–15) | 805 | 36.5 | 31.4–41.9 | 936 | 47.3 | 42.1–52.5 | 1 855 | 63.9 | 59.7–67.8 | 1 473 | 79.9 | 76.4–82.9 | 43.4 | <0.001 |
| High risk/harmful (16–19) | 143 | 33.4 | 24.5–43.6 | 120 | 37.4 | 25.2–51.4 | 340 | 62.1 | 53.6–70.0 | 253 | 75.7 | 66.7–83.0 | 42.3 | <0.001 |
| High risk (20+) | 1 124 | 38.6 | 34.3–43.0 | 128 | 50.2 | 37.4–63.1 | 295 | 65.7 | 56.3–73.9 | 263 | 71.5 | 62.5–79.0 | 32.9 | <0.001 |
| **Correct HIV knowledge and myth rejection** | | | | | | | | | | | | | | |
| No knowledge | 9 184 | 28.4 | 26.7–30.2 | 8 902 | 48.3 | 46.7–49.9 | 18 716 | 64.4 | 62.9–65.8 | 14 652 | 74.1 | 72.7–75.4 | 45.7 | <0.001 |
| Yes knowledge | 6 904 | 33.4 | 31.3–35.6 | 4 138 | 55.0 | 52.4–57.4 | 7 555 | 68.9 | 67.0–70.7 | 8 492 | 77.3 | 75.7–78.9 | 43.9 | <0.001 |
| **Self–perceived risk of HIV infection** | | | | | | | | | | | | | | |
| No risk | 11 151 | 28.6 | 26.8–30.5 | 10 081 | 48.0 | 46.3–49.6 | 4 904 | 75.1 | 73.1–77.1 | 18 383 | 72.3 | 70.9–73.7 | 43.7 | <0.001 |

*(Continued)*

**Table 2.** (Continued)

| Variables | 2005 | | | 2008 | | | 2012 | | | 2017 | | | Percentage point increase | p-value* |
|---|---|---|---|---|---|---|---|---|---|---|---|---|---|---|
| | n | % | 95% CI | n | % | 95% CI | n | % | 95% CI | n | % | 95% CI | | |
| Yes risk | 4 914 | 34.1 | 31.9–36.3 | 2 923 | 56.2 | 53.7–58.8 | 21 240 | 62.3 | 60.8–63.8 | 2 836 | 79.3 | 76.8–81.5 | 45.2 | <0.001 |
| **HIV status** | | | | | | | | | | | | | | |
| HIV Negative | 10 510 | 29.3 | 27.6–31.0 | 8 929 | 48.2 | 46.5–49.8 | 17 872 | 62.3 | 60.8–63.8 | 13 193 | 74.7 | 73.4–75.9 | 45.4 | <0.001 |
| HIV Positive | 1 328 | 36.7 | 32.9–40.7 | 1 227 | 65.2 | 61.2–69.0 | 2 605 | 81.7 | 79.2–84.0 | 2 604 | 87.1 | 84.7–89.2 | 50.4 | <0.001 |

* p-value ≤ 0.05 was considered statistically significant

HIV than those not employed, in 2005 [aOR = 1.50 (95% CI: 1.27–1.78), p<0.001], 2008 [aOR = 1.55 (95% CI: 1.30–1.83), p<0.001], 2012 [aOR = 1.3 (95% CI: 1.2–1.5), p<0.001], and 2017 [aOR = 1.34 (95% CI: 1.22–1.48, p<0.001]. Those with accurate knowledge of preventing the sexual transmission of HIV and rejection of misconceptions about HIV transmission were significantly more likely to have ever tested for HIV, in 2005 [aOR = 1.17 (95% CI: 1.00–1.36), p = 0.052], 2012 [aOR = 1.2 (95% CI: 1.1–1.4), p = 0.006], and 2017[aOR = 1.15 (95% CI: 1.05–1.25), p = 0.002]. The respondents who perceived themselves as being at risk of HIV infection were significantly more likely to have ever tested for HIV, in 2005 [aOR = 1.23 (95% CI: 1.05–1.44), p = 0.009], 2008 [aOR = 1.30 (95% CI: 1.10–1.52), p = 0.002], and 2017 [aOR = 1.40

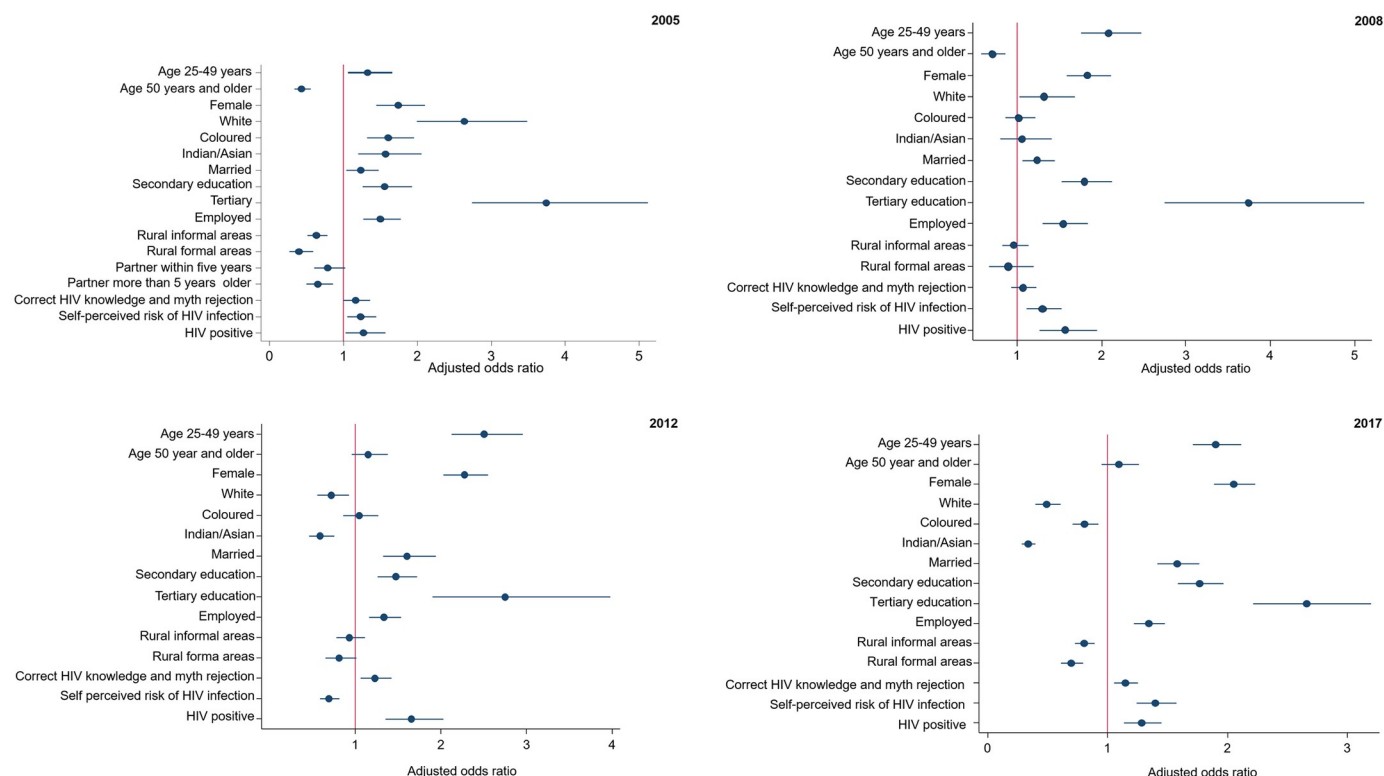

**Fig 1. Multivariate logistic regression models of determinates of having ever been tested for HIV in South Africa by survey period from 2005–2017.**

(95% CI: 1.24–1.57), p<0.001], except in 2012 [aOR = 0.7 (95% CI: 0.6–0.8), p<0.001]. Those who tested HIV positive were significantly more likely to have ever tested for HIV, in 2005 [aOR = 1.27 (95% CI: 1.03–1.57), p = 0.027], 2008 [aOR = 1.57 (95% CI: 1.27–1.94), p<0.001], and 2012 [aOR = 1.7 (95% CI: 1.4–2.0), p<0.001], and in 2017 [aOR = 1.28 (95% CI: 1.13–1.54), p<0.001].

The decreased likelihood of being tested for HIV was significantly associated with those aged 50 years and older years than those aged 15–24 years, in 2005 [aOR = 0.43 (95% CI: 0.34–0.56), p<0.001], and 2008[aOR = 0.70 (95% CI: 0.57–0.86), p<0.001]. The decreased likelihood of ever having tested for HIV was significantly associated with being white in 2012 [aOR = 0.7 (95% CI: 0.6–0.9), p = 0.011], and 2017 [aOR = 0.49 (95% CI: 0.40–0.61), p<0.001] than Black Africans. In 2012, Indians/Asians were significantly less likely to ever have tested for HIV than Black Africans [aOR = 0.34 (95% CI: 0.28–0.40), p<0.001] and in 2017 [aOR = 0.34 (95% CI: 0.28–0.40), p<0.001]. Coloureds were significantly less likely to ever have tested for HIV than Black Africans in 2017 [aOR = 0.81 (95% CI: 0.71–0.92), p<0.001]. Those residing in rural informal areas were significantly less likely to have ever have tested for HIV in 2005 [aOR = 0.63(95% CI: 0.51–0.78), p<0.001], and 2017 [aOR = 0.80 (95% CI: 0.73–0.89), p<0.001] including those residing in rural formal areas in 2017 [aOR = 0.70 (95% CI: 0.61–0.79), p<0.001] compared to respondents from urban areas.

## Discussion

This study reports results on HIV testing trends for those who have ever tested across four nationally representative surveys in the South African population aged 15 years and older.

The observed overall significant increase in youth and adults 15 years and older who have ever been tested for HIV is encouraging, especially among Black Africans. This suggests that the national HIV testing programmes are gradually reaching these two groups that are still the most vulnerable to HIV infection. There is also an indication that over time (between the study waves) the national HTS programme is reaching high-risk groups, especially those who are sexually active, those with older sexual partners and those who tested positive in the survey rounds. Scaling up access and outreach to testing among most at risk populations remains an important goal for universal access to treatment and support in South Africa [20].

Understanding the associations between ever having tested for HIV and these identified factors may help improve HIV testing policy and increase testing utilization, which together will lead to better control of the HIV epidemic in South Africa. The current findings revealed that over the past decade people aged 25–49 years were more likely to test for HIV than people aged 15–24 years. These findings are similar to the findings from other studies [21, 22]. This is not surprising, as older people would have had more opportunities to have ever tested compared to those who are younger. What is concerning is that younger people represent the largest proportion of new HIV cases in South Africa [15]. Providing increased access to HIV testing in educational settings will increase awareness of HIV status among the youth [23]. Studies have also shown that by including youth in the planning and development of HIV testing strategies and by providing youth friendly environments for HIV testing, increases the uptake of HIV testing in this age group [23, 24].

The findings also show that the proportion of people who have ever tested for HIV was lower among those residing in rural areas compared to residing in urban areas [25]. This could reflect poor programme outreach in rural areas due to the focus on urban areas as a result of the high prevalence of HIV in the urban areas. This disparity in service delivery is well documented in other studies [26]. There is need to expand services in rural areas in order to increase access to testing services for the rural population. Community-based approaches such

as mobile health clinics and door-to-door campaigns have been shown to be successful in reaching populations that do not present at health facilities and where there are inadequate fixed facilities to provide HIV testing services [27–29].

The findings showed that HIV testing has increased substantially among Black Africans. This may be due to the targeted and/or focussed national testing efforts. This may reflect the success of the national HIV testing and counselling campaign launched in 2010, and the revitalised HIV testing services which focused on getting more people to test for HIV [8, 20]. However, there is a growing concern with regard to the Coloured race group, as there seems to be a brewing HIV epidemic among this population [15]. The changing racial dynamics over time suggest that efforts to get people to test have been uneven and need to be addressed. HIV testing services therefore need to be inclusive to prevent the oversight of other racial groups in providing this critical service across the board [30]. The proportion of Whites and Indian/ Asians ever having tested for HIV increased like every other race group but not to the same extent, a 16.2% and 17.1% point increase respectively as compared to a 37.6% point increase for Coloureds and a 50.3% point increase for Black Africans. Previous research shows that other race groups are often reluctant to test, as they perceive themselves as not being at risk of HIV [14, 15].

In addition, the findings showed that marriage, education, and employment is associated with an increase in HIV testing. This is similar to other studies [31–33]. In South Africa, associations have also been shown between an individual's socio-economic background and the likelihood to test for HIV [14, 15]. The findings indicate that there is a need to target people with no or little formal education and those not employed.

These findings also showed that those who tested positive in the survey rounds were more like to have tested for HIV. This is encouraging as awareness of HIV status among those who are HIV positive is essential to ensure that people living with HIV are supported and receive treatment.

## Limitations

Some key limitations for our study include that data on HIV testing, socio-demographic and HIV related risk behaviours were collected using self-reports, and so were subject to social desirability and recall biases. Furthermore, each survey wave was cross-sectional in nature rather than longitudinal. The study is therefore limited to assessing the associations between HIV testing and potential determinants as one cannot infer causality among the variables studied. Nevertheless, the study provides nationally representative data on trends in ever testing for HIV that can be inferred to the general youth and adult population in South Africa.

## Conclusion

It is encouraging that HIV testing uptake has increased significantly from 2005 to 2017 in South Africa. HIV testing programmes have reached those most at risk of HIV infection. However, more HIV testing opportunities in different settings that prompts both provider and client initiated approaches are required to ensure that low HIV testing groups are reached. The findings inform the need for a comprehensive strategy targeting young people, those living in rural areas, the never married, those with no formal education or low educational attainment, and the unemployed. Box 1 outline key challenges identified and recommendations for improving HIV testing.

Box 1. Key challenges to HIV testing and policy recommendations.

| Key challenges | Policy recommendations |
|---|---|
| Low HIV testing among young people 15–24 years | • Educating the youth to increase awareness and reduce the fear surrounding testing<br>• Including youth in the planning and development of HIV testing strategies<br>• Providing youth friendly environments for HIV testing |
| Lower HIV testing among minority race groups | • There is a need for development of policies and practices meant to reduce racial disparities in HIV testing<br> o Targeted measures against stigmatization, discrimination, and mistrust should be developed and deployed to enhance HIV testing minority race |
| Lower HIV testing among those who never married | • HTS should be reinforced at community level<br> o Home-based voluntary counselling and testing service is highly recommended to increase uptake of HTS<br> o Reducing community level stigma should be a priority towards improving HIV testing uptake |
| Lower HIV testing among those residing in rural areas | • Expanding services in rural areas in order to increase access to testing services for the rural population<br> o Through community based approaches<br> ▪ Mobile testing units and door-to-door campaigns |
| Lower HIV testing among those with lower educational attainment | • Addressing the educational needs in the country is crucial among predominantly lower educated groups<br>• Providing initiatives to overcome barriers to testing among those with no formal education or lower levels of educational attainment |
| Lower HIV testing among the unemployed | • There is a need for HIV counselling and testing campaigns targeting the unemployed<br>• Steps are also needed to facilitate healthcare access and ensure the rapid delivery of HIV testing among the unemployed<br>• Research is needed as to how interventions should provide tailored support for the unemployed |

## Supporting information

**S1 Data.**
(ZIP)

## Acknowledgments

We would like to thank study participants who willingly opened their doors and hearts to give us private information about themselves for the sake of contributing to a national effort to contain the spread of HIV. We are grateful to our international partners and our research consortium members for unwavering support. Finally yet importantly, we would like to thank the project team that made this survey a success.

## Author Contributions

**Conceptualization:** Sean Jooste.

**Formal analysis:** Sean Jooste, Musawenkosi Mabaso.

**Writing – original draft:** Sean Jooste.

**Writing – review & editing:** Musawenkosi Mabaso, Myra Taylor, Alicia North, Rebecca Tadokera, Leickness Simbayi.

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
