## [Decision Letter · Decision Letter 0]

8 Jan 2020

PONE-D-19-23579

Trends and determinants of ever having tested for HIV among youth and adults in South Africa from 2005–2017: Results from four repeated cross-sectional nationally representative household-based HIV Prevalence, Incidence, and Behaviour surveys

PLOS ONE

Dear Mr Jooste,

Thank you for submitting your manuscript to PLOS ONE. After careful consideration, we feel that it has merit but does not fully meet PLOS ONE’s publication criteria as it currently stands. Therefore, we invite you to submit a revised version of the manuscript that addresses the points raised during the review process.

We would appreciate receiving your revised manuscript by Feb 22 2020 11:59PM. To enhance the reproducibility of your results, we recommend that if applicable you deposit your laboratory protocols in protocols.io, where a protocol can be assigned its own identifier (DOI) such that it can be cited independently in the future. For instructions see: http://journals.plos.org/plosone/s/submission-guidelines#loc-laboratory-protocols

We look forward to receiving your revised manuscript.

Kind regards,

Florian Fischer

Academic Editor

PLOS ONE

Journal Requirements:

'This paper has been supported by the President’s Emergency Plan for AIDS Relief (PEPFAR) through the CDC under the terms of 5U2GGH000570, U2GGH000357, and NU2GGH001629. Its contents a 348 re solely the responsibility of the authors and do not necessarily represent the official views of CDC.  Financial support was received from the Nelson Mandela Foundation and the Swiss Agency for Development and Cooperation for 2005 survey. Bill & Melinda Gates Foundation provided support for the 2012, while the United Nations Children’s Fund (UNICEF) funded the 2008, 2012 and 2017 survey. The South African National AIDS Council (SANAC), United States Agency for International Development (USAID), Soul City; LoveLife, the Centre for Communication Impact (CCI) provided funding for the 2017 survey.'

Please include in your financial disclosure statement the name of the funders of this study (as well as grant numbers if available). At present, this information is only available in your acknowledgement section.

'The funders had no role in study design, data collection and analysis, decision to publish, or preparation of the manuscript.'

Please provide an amended Funding Statement that declares *all* the funding or sources of support received during this specific study (whether external or internal to your organization) as detailed online in our guide for authors at http://journals.plos.org/plosone/s/submit-nowPlease state what role the funders took in the study.  If any authors received a salary from any of your funders, please state which authors and which funder. If the funders had no role, please state: "The funders had no role in study design, data collection and analysis, decision to publish, or preparation of the manuscript."

3. Please ensure that your references are formatted according to the PLOS ONE submission guidelines https://journals.plos.org/plosone/s/submission-guidelines#loc-references. In particular, please note that references should be listed at the end of the manuscript and numbered in the order that they appear in the text. In the text, please cite the reference number in square brackets.

4. Please provide additional details regarding participant consent for the surveys and the collection of the dried blood spot specimens.

In the ethics statement in the Methods and online submission information, please ensure that you have specified (i) whether consent was informed and (ii) what type you obtained (for instance, written or verbal, and if verbal, how it was documented and witnessed).

As your study included minors, please also state whether you obtained consent from parents or guardians.

Reviewers' comments:

Reviewer's Responses to Questions

**Comments to the Author**

1. Is the manuscript technically sound, and do the data support the conclusions?

Reviewer #1: Yes

Reviewer #2: Yes

2. Has the statistical analysis been performed appropriately and rigorously? 

Reviewer #1: Yes

Reviewer #2: Yes

3. Have the authors made all data underlying the findings in their manuscript fully available?

Reviewer #1: Yes

Reviewer #2: Yes

4. Is the manuscript presented in an intelligible fashion and written in standard English?

Reviewer #1: Yes

Reviewer #2: Yes

5. Review Comments to the Author

Reviewer #1: The work from this group and the national analyses they have conducted have been extremely crucial to understanding the HIV epidemic in South Africa and the impact of health policy changes on the HIV care cascade. While the data presented is regionally specific, the trends they demonstrate likely mirror other parts of sub Saharan Africa and thus the study may be of interest as it can likely be reproduced in other countries to develop a longitudinal understanding.

While the article is well written and the data are robust, I strongly believe that the paper can be substantially be improved by the addition of the following suggestion and an overall reframing on how this analysis can help us understand the impact of national health policy and make recommendations for future health policy priorities. Please see my specific comments below.

Introduction: In paragraph two the authors present a very generic discussion around barriers to testing and linkage to care. They further discuss that, “Although some studies found that individual factors affect uptake of HIV testing, some found mixed results that included factors relating to community, society and relationships’ type” This argument and sentence feels somewhat mis-placed since statics cross sectional population based surveys are unlikely to provide the depth to understand these associations.

What is missing and necessary is two things:

1. How has health policy around HIV testing, and LTC changed in South Africa for the last fifteen years - these authors are experts and should be able to pain the picture.

2. What is the benefit of this type of longitudinal analysis, has these been done elsewhere , has it changed policy, why did the authors write this paper?

Methods: Nil to add, well written and thorough

Results:

Table 1 and 2, are necessary and complete

Couple of comments:

1. The paragraphs under determinants of health is extremely dense and difficult to read and thus asorb. May be the authors should draw a conceptual model with the demographic and behavioral factors tested, with some color distinction for those with stronger associations and those with weaker associations. This would be more meaningful, than the paragraphs written.

2. While I like figure 1, it is too much data for the reader to absorb, also the figure need s to be edited – font/spacing etc so that it is more readable on a web-based platform.

3. I would love to see a graphical representation of the care cascade over time, I would also like to see and interrupted time-series analysis that high lights temporal impact of national policy roles outs and on the subsequent household survey. This should be before tables 1 and 2, and should show the big picture.

Discussion:

While the discussion is well written it could be further strengthened. The authors are presenting really rich data, and so should end the discussion with key policy recommendations. Maybe this is a summary box, that ties together the key findings from each paragraph.

Reviewer #2: This manuscript is well written and provides a descriptive analysis of trends and associations in ever HIV testing in South Africa.

Introduction

The introduction provides a sound rational for the study/analysis questions. The authors describe the multiple layers (individual, social, and structural) that influence HIV testing. Would consider grounding it in the eco-social framework for HIV transmission (Baral et al BMC Public Health 2013).

Methods

The methodology was well described, however it would be helpful to have additional information on the adjustments made for the complex survey analysis, and how missing data was handled.

The authors described DBS for HIV testing, but these results were not presented in the manuscript. If they were not used in the analysis, then I would limit the description of the HIV testing to a procedure that was done as part of the survey, but data from testing was not presented.

Results

The results were well described and the tables were easy to read.

For the multivariate analysis, all the odds ratios should be adjusted odds ratios given that there were multiple variables used for adjustment.

Minor point - there should be a space between 95% CI, it's missing for several lines in this part of the results.

Figure 1 is very blurry. It would also be helpful to included "Education" for "Secondary" and "Tertiary" for the model.

6. PLOS authors have the option to publish the peer review history of their article (what does this mean?). If published, this will include your full peer review and any attached files.

Reviewer #1: Yes: b Hansoti

Reviewer #2: No

---

## [Author Response · Author response to Decision Letter 0]

21 Feb 2020

Response: The manuscript has been formatted according to journal guidelines.

Please include in your financial disclosure statement the name of the funders of this study (as well as grant numbers if available). At present, this information is only available in your acknowledgement section.

Response: Financial disclosure has now been moved from acknowledgements to the Funding Statement. 

'The funders had no role in study design, data collection and analysis, decision to publish, or preparation of the manuscript.'

a. Please provide an amended Funding Statement that declares *all* the funding or sources of support received during this specific study (whether external or internal to your organization) as detailed online in our guide for authors at http://journals.plos.org/plosone/s/submit-now

b. Please state what role the funders took in the study. If any authors received a salary from any of your funders, please state which authors and which funder. If the funders had no role, please state: "The funders had no role in study design, data collection and analysis, decision to publish, or preparation of the manuscript."

Response: The funding statement has also been updated as follows 'The funders had no role in study design, data collection and analysis, decision to publish, or preparation of the manuscript. The cover latter has also been updated as suggested.

3. Please ensure that your references are formatted according to the PLOS ONE submission guidelines https://journals.plos.org/plosone/s/submission-guidelines#loc-references. In particular, please note that references should be listed at the end of the manuscript and numbered in the order that they appear in the text. In the text, please cite the reference number in square brackets.

Response: The references have been formatted according to journal guidelines.

4. Please provide additional details regarding participant consent for the surveys and the collection of the dried blood spot specimens.

In the ethics statement in the Methods and online submission information, please ensure that you have specified (i) whether consent was informed and (ii) what type you obtained (for instance, written or verbal, and if verbal, how it was documented and witnessed).

As your study included minors, please also state whether you obtained consent from parents or guardians.

Response: Written Informed consent for participating in the survey and for the collection of the dried blood spot specimens were obtained or participants 18 years and older. For those less than 18 years informed consent was obtained from parents or guardians and assent obtained from the participants. Where people could not write verbal consent was obtained and fieldwork supervisor signed as witness.

Response: Data used in this analysis will be made available as supporting information.

5. Review Comments to the Author

Introduction: In paragraph two the authors present a very generic discussion around barriers to testing and linkage to care. They further discuss that, “Although some studies found that individual factors affect uptake of HIV testing, some found mixed results that included factors relating to community, society and relationships’ type” This argument and sentence feels somewhat mis-placed since statics cross sectional population based surveys are unlikely to provide the depth to understand these associations.

Response: The line has been deleted and paragraph revised, see line 86-96. 

What is missing and necessary is two things:

1. How has health policy around HIV testing, and LTC changed in South Africa for the last fifteen years - these authors are experts and should be able to pain the picture.

Response: This is now captured in a new paragraph in lines 94-98.

2. What is the benefit of this type of longitudinal analysis, has these been done elsewhere , has it changed policy, why did the authors write this paper?

Response: This is now captured in a new paragraph in lines 99-102.

Methods: Nil to add, well written and thorough

Results:

Table 1 and 2, are necessary and complete

Couple of comments:

1. The paragraphs under determinants of health is extremely dense and difficult to read and thus asorb. May be the authors should draw a conceptual model with the demographic and behavioral factors tested, with some color distinction for those with stronger associations and those with weaker associations. This would be more meaningful, than the paragraphs written.

Response: Unfortunately most readers would like to see results in the text and would like to keep them. 

2. While I like figure 1, it is too much data for the reader to absorb, also the figure need s to be edited – font/spacing etc so that it is more readable on a web-based platform.

Response: The figure has been greatly improved, and has been put through PACE.

3. I would love to see a graphical representation of the care cascade over time, I would also like to see and interrupted time-series analysis that high lights temporal impact of national policy roles outs and on the subsequent household survey. This should be before tables 1 and 2, and should show the big picture.

Response: The focus of this analysis was not on the care cascade but rather on HIV testing overtime as it affects the UNAIDS indicators, especially the first 90. Besides there is no data on temporal impact of national policy roles outs and on the subsequent household survey.

Discussion:

While the discussion is well written it could be further strengthened. The authors are presenting really rich data, and so should end the discussion with key policy recommendations. Maybe this is a summary box, that ties together the key findings from each paragraph.

Response: Key challenges and recommendations are now included, see lines 355-364

Reviewer #2: This manuscript is well written and provides a descriptive analysis of trends and associations in ever HIV testing in South Africa.

Introduction

The introduction provides a sound rational for the study/analysis questions. The authors describe the multiple layers (individual, social, and structural) that influence HIV testing. Would consider grounding it in the eco-social framework for HIV transmission (Baral et al BMC Public Health 2013).

Response: Revised, see line 86-98

Methods

The methodology was well described, however it would be helpful to have additional information on the adjustments made for the complex survey analysis, and how missing data was handled.

Response: Sample weights were introduced to all models account for the complex survey design and non-response using the ‘svy’ command, see lines 181-183.

The authors described DBS for HIV testing, but these results were not presented in the manuscript. If they were not used in the analysis, then I would limit the description of the HIV testing to a procedure that was done as part of the survey, but data from testing was not presented.

Response: Antibody detected HIV status (HIV positive, HIV negative) was included in the analysis as covariate, see line 170.

Results

The results were well described and the tables were easy to read.

For the multivariate analysis, all the odds ratios should be adjusted odds ratios given that there were multiple variables used for adjustment.

Response: Corrected as suggested.

Minor point - there should be a space between 95% CI, it's missing for several lines in this part of the results.

Response: Corrected as suggested.

Figure 1 is very blurry. It would also be helpful to included "Education" for "Secondary" and "Tertiary" for the model.

Response: The figure has been greatly improved as suggested. The figure has been put through PACE.

---

## [Decision Letter · Decision Letter 1]

24 Apr 2020

Trends and determinants of ever having tested for HIV among youth and adults in South Africa from 2005–2017: Results from four repeated cross-sectional nationally representative household-based HIV Prevalence, Incidence, and Behaviour surveys

PONE-D-19-23579R1

Dear Dr. Jooste,

We are pleased to inform you that your manuscript has been judged scientifically suitable for publication and will be formally accepted for publication once it complies with all outstanding technical requirements.

With kind regards,

Florian Fischer

Academic Editor

PLOS ONE

Additional Editor Comments (optional):

Reviewers' comments:

Reviewer's Responses to Questions

**Comments to the Author**

1. If the authors have adequately addressed your comments raised in a previous round of review and you feel that this manuscript is now acceptable for publication, you may indicate that here to bypass the “Comments to the Author” section, enter your conflict of interest statement in the “Confidential to Editor” section, and submit your "Accept" recommendation.

Reviewer #1: All comments have been addressed

2. Is the manuscript technically sound, and do the data support the conclusions?

Reviewer #1: Yes

3. Has the statistical analysis been performed appropriately and rigorously? 

Reviewer #1: Yes

4. Have the authors made all data underlying the findings in their manuscript fully available?

Reviewer #1: Yes

5. Is the manuscript presented in an intelligible fashion and written in standard English?

Reviewer #1: Yes

6. Review Comments to the Author

Reviewer #1: The manuscript is greatly improved thank you for making the suggested edits and modifying the discussion as suggested.

7. PLOS authors have the option to publish the peer review history of their article (what does this mean?). If published, this will include your full peer review and any attached files.

Reviewer #1: No

---

## [Editor Report · Acceptance letter]

1 May 2020

PONE-D-19-23579R1 

Trends and determinants of ever having tested for HIV among youth and adults in South Africa from 2005–2017: Results from four repeated cross-sectional nationally representative household-based HIV Prevalence, Incidence, and Behaviour surveys 

Dear Dr. Jooste:

I am pleased to inform you that your manuscript has been deemed suitable for publication in PLOS ONE. Congratulations! Your manuscript is now with our production department. 

With kind regards,

on behalf of

Dr. Florian Fischer 

Academic Editor

PLOS ONE